# Measuring Sheep Tails: A Preliminary Study Using Length (Mm), Vulva Cover Assessment, and Number of Tail Joints

**DOI:** 10.3390/ani13060963

**Published:** 2023-03-07

**Authors:** Madeleine Woodruff, Carolina Munoz, Grahame Coleman, Rebecca Doyle, Stuart Barber

**Affiliations:** 1The Animal Welfare Science Centre, Faculty of Science, The University of Melbourne, Melbourne, VIC 3052, Australia; 2Faculty of Science, Melbourne Veterinary School, The University of Melbourne, Melbourne, VIC 3052, Australia; 3Jeanne Marchig International Centre for Animal Welfare Education, The Royal (Dick) School of Veterinary Studies, University of Edinburgh, Edinburgh EH9 1RS, UK

**Keywords:** tail, best practice, welfare, length, monitoring, merino

## Abstract

**Simple Summary:**

It is common for sheep to have tails docked as lambs, predominantly to reduce flystrike. It is recommended that tails are docked long enough to cover the vulva in ewes and at a similar length in males. There are many ways that tail length in sheep is described and measured, but there is a lack of congruency across the different description and measurement methods. This study aimed to investigate the reliability of three common ways to measure and describe docked sheep tail length, using 99 Merino ewes, 51 yearlings (1.4-year-olds) and 48 weaners (5-month-olds). Length and vulva cover assessment were the methods with the most reliable results, joint palpation was the least reliable method of tail measurement. The tails in this sample that were of the recommended length, covering the vulva, measured 10.8–16.2 mm longer than tails not covering the vulva, at 63.7 mm for weaners and 57.6 mm for yearlings on average, and contained more than two coccygeal vertebrae. This research presents two reliable methods of sheep tail measurement for two age groups and provides a foundation for future research into method refinement and tail length monitoring.

**Abstract:**

Docking sheep tails is a long-standing practice that, when done at the recommended length, reduces the risk of flystrike. The recommended length is to cover the vulva of ewes and to a similar length in males. This length is often equated to three coccygeal joints left intact, and there are many other ways the recommended length is described by researchers, industry, and government. This study compared the observer consistency and retest consistency using three different tail length measurement methods: vulva cover assessment, length (mm), and joint palpation. The tails of 51 yearling and 48 weaner Merino ewes were assessed by two observers. Length and vulva cover assessment methods provided the most reliable results, and joint palpation was the least reliable method of tail measurement. In the sample, tails that covered the vulva of yearlings and weaners measured 57.6 mm (*n* = 14) and 63.7 mm (*n* = 30) on average, respectively, and contained two coccygeal joints (more than two coccygeal vertebrae). Tails that did not cover the vulva of yearlings and weaners measured 41.3 mm (*n* = 36) and 52.8 mm (*n* = 17) on average, respectively, and had less than two coccygeal joints. The two most reliable methods enable valid comparison to the best practice recommendations.

## 1. Introduction

Docking the tails of sheep is a very common animal husbandry practice, particularly in countries that manage sheep in extensive systems such as Australia [1,2,3,4]. European archaeological and historical research indicates that the practice began in the late Middle Ages, likely in the belief it might maintain a clean breech, aid mating and parturition, and it was perceived to create the appearance of an aesthetically rounded rump [5]. Many of these reasons for tail docking have persisted [2,3]. The prevention of flystrike (cutaneous myiasis) and breech cleanliness are the primary reasons producers dock sheep tails in Australia [6] and around the world [1,3]. Flystrike is a significant welfare issue and an Australian priority endemic disease, costing the industry AUD 300 million in treatment, prevention, and production losses annually [7].

Research into the optimal length to dock tails to prevent flystrike was conducted in Australia from the 1930s. The outcome of this research was the recommendation to dock tails long enough to cover the vulva in ewes, which approximately equated to leaving three joints of the tail [8,9,10,11,12]. Compared with tails docked shorter, the recommended length reduced the risk of cancer [12,13], rectal prolapse [14,15], and bacterial arthritis [16] and was considered to be a practical length for on-farm practices such as shearing and crutching [17]. Therefore, sheep with tails docked shorter than recommended are considered to have compromised welfare due to the increased risk of these health outcomes. The recommended sheep tail docking length has been included in the Australian Codes of Farming Practice [18,19] which preceded the National Animal Welfare Standards and Guidelines [20], industry practice handbooks and magazines [21,22], and state agriculture department websites [23]. However, among the scientific and other publications, there is no one way that tail docking length is described. Common ways of describing tail length, dating back to the early scientific literature, include the level of cover of the vulva and/or anus [11,12], length (centimetres/inches) measurement [11,24], the number of palpable coccygeal joints [24,25,26], and/or the relationship to the caudal fold attachments [15,27,28]. Industry and government publications use a combination of these descriptions to portray the recommended length [21,22,23]. These various descriptions may present challenges such as communicating and understanding the recommended tail docking length, interpreting producers’ self-reported practice, and assessing tail length on-farm. These challenges potentially contribute to the proportion of between 24–54% of sheep with tails docked shorter than recommended, as reported by producers over the past two decades in the Australian flock [29,30,31,32]. Woodruff et al. (2020) found some producers’ descriptions of sheep tail docking length lacked acknowledgement of vulva coverage and/or used multiple descriptions that appeared contradictory [6]. The challenges around interpreting self-reported data and assessing on-farm tail length complicate research into current practice and adherence to best practice which, in turn, impacts the industry’s ability to address the issue effectively and improve the welfare of the Australian flock.

Although there have been studies that have measured tail length, there is limited consistency in methods, results, and reliability, where reported. Seven publications describe experimental studies that use vulva cover [11,24,25], palpable coccygeal joints [24,25,33,34], caudal folds [27,28], and/or measured length (cm/inches) [11,24,25,27,28,33,34] to describe tail length at docking and other age points. It has been demonstrated that tails docked just below the tip of the vulva [11], with the caudal folds intact [27,28] or through the third coccygeal joint space [33], when assessed between weaning age and 10 months old measured between 40.9 and 71 mm in total length on average [27,28,33], and extended 25 mm beyond the tip of the vulva [11]. In sheep that had tails docked shorter than recommended through the first and second joint space [33], or with the vulva completely or half exposed [11], when assessed at eight to nine months old these tails measured 23 mm, 42.1 mm [33], and 9.5 mm above and 9.5 mm below the tip of the vulva [11] on average, respectively. In two studies, sheep that had tails docked through the third joint space [33] or below the tip of the vulva [11] were followed through to around two years old, their tails measured 71.8 mm [33], and extended 19.1 mm below the tip of the vulva [11] on average. Whereas sheep that had tails docked through the first or second joint space measured 17.7 mm and 38.5 mm on average, respectively, when assessed again at around 2 years old [33]. There is wide range in how tails were measured, ages of sheep assessed, sample sizes, and therefore, results. The management and breeds of sheep in the studies conducted in the United States of America [27,28] differ greatly from that of Australian systems, where docking age was significantly younger than current Australian practice and previous Australian tail length research. Just two of the common tail length measurement methods used in research reported analysing reliability. These studies investigated on-farm welfare measures and demonstrated a range of poor to perfect reliability between and/or within observers using the vulva cover method using the Weighted Kappa statistic [35,36], and high percentage agreement [35]. One study assessed reliability of tail length (cm), and reported a strong correlation between observers using a device created to standardise the measurement [27]. Not one study used all or most methods and results varied across the studies leading to difficulty in interpreting results and making direct comparisons across studies and producing clear guidelines. There is a lack of information about what tools and methods were used to measure tails and this is necessary so that these methods can be utilised by producers and/or researchers on-farm. While these studies enable comparisons to be made between some of the common tail length measurement methods, there remains a need to achieve more general agreement.

Monitoring how effective a practice is, and how it compares to best practice, provides producers with information for change and improvement. There are many monitoring measures of sheep health, welfare, and productivity on-farm including assessing body-weight, assessing pasture quantity and nutritional quality [37], body condition scoring [35,37], lameness and dag scoring [35], and worm egg counts [29,38]. Producers can utilise the information to inform their management and make adjustments as necessary [35,37]. These monitoring measures occur at regular time points throughout the sheep’s life; for example, it is recommended by industry to collect worm egg counts seasonally for the assessment of parasite burden, drench requirements, and superior genotypes for breeding [38]. Tail docking, however, occurs once for lambs and there is little information on whether producers assess the docked tail length of their sheep throughout the life of the sheep or assess for variation. As previously mentioned, in Australia, Graham, Johnstone, and Riches (1947) demonstrated that tails docked just below the tip of the vulva, maintained coverage of the vulva when followed up at eight months and two years of age, on average. It was also observed that some tails docked to partially cover the vulva grew beyond the tip of the vulva at follow-up ages (four and a half months, eight months, and two years after docking). However, it is unclear how tails were measured, what the level of operator variation at docking or assessment was, and what the follow-up length appearance was compared against, whether it was photographs or written descriptions, for example. It was noted that tails docked to completely or partially cover the vulva were more commonly docked too short rather than too long [11]. In the United States of America, Goodwin et al. (2007) and Lewis (2013) found that tails docked through or after the caudal folds increase by 8.1 mm [27] and 26 mm [28] from docking to weaning, and by 4.2 mm from weaning to market [27], on average. While these studies have shown that tails increase in length after docking [27,28], and that docked tails can maintain or achieve vulva coverage [11] as the sheep matures, tails that are docked too short may never extend to or past the tip of the vulva, leading to lasting impacts over a ewe’s lifetime. For sheep producers, having ways to assess tail length practically and reliably on-farm may enable monitoring and planning for adjustments for the next yearly lamb tail docking.

The current study posed the questions: ‘What are the relationships between different ways of describing sheep tail length?’ and ‘What is or are the ideal method(s) for measuring sheep tail length on-farm?’ 

## 2. Materials and Methods

### 2.1. General Study Information

This study was approved by The University of Melbourne Animal Ethics Committee (Ethics ID: 20246) and was carried out on a commercial Merino sheep farm in January 2021 on Yorta Yorta Country in Northern Victoria, Australia. The commercial Merino sheep farm had more than 2000 breeding ewes managed extensively. A total of 99 animals were assessed consistent with the recommendations in the Animal Welfare Indictors (AWIN) protocol for assessing sheep welfare for farms with 2000 breeding ewes or more [39]. The study was conducted over two consecutive days. On Day 1, an assessment of tail length was conducted by two assessors (further details of the tail assessment are provided below), on Day 2, a re-assessment of tail length was done by the same assessors. The timing of this study visit was organised to fit in with other on-farm livestock activities. All sheep were identified via electronic National Livestock Identification Scheme (NLIS) ear tags placed on lambs at tail docking, prior to weaning.

### 2.2. The Sheep

This study assessed the tails of 99 unmulesed Merino sheep. Five Merino yearlings were used solely for observer training at the beginning of the trial. In total, 48 weaner and 51 yearling ewes had tails assessed. All sheep had been tail docked with Elastrator^®^ rings using a Numnuts^®^ applicator (www.numnuts.store) that combines the application of the ring with an injection of lignocaine to the tail. The yearlings (14–15 months old) were born in August 2019 (over a 4 week lambing period), tail docked in the first week of September 2019, and were shorn in October 2020. The weaners (4–5 months old) were born in August 2020 (over a 4-week lambing period), tail docked in the first week of September 2020, and were crutched in December 2020.

### 2.3. The Measurement Methods

The three methods of tail length measurement used were vulva cover, joint palpation, and length in millimetres (mm) measurement, further described in Table 1. The order of assessment was as follows: (1) Vulva cover, (2) Joint palpation, (3) Length. In addition, one infrared (IR) image per sheep was collected (Day 1). Body condition score (BCS) of yearling ewes only were assessed, and all sheep were weighed on Day 2. All assessments took place with the sheep standing in the race.

### 2.4. Infrared Image

There are no results from the infrared images as they were not suitable for analysis for this paper; however, presented are the methods used. Using a Flir E4 infrared camera, a dual thermal photographic image was captured. This method was used to assess temperature differences between exposed perineal bare skin and the tail. This method was included as a potential remote alternative to assess vulva cover. The camera was used approximately 1 m from the tail, Observer 1 crouched down behind the sheep for a tail-level view and wool that hung down over the end of the tail (yearlings) was gently lifted up to allow view of the end of the tail or vulva. 

### 2.5. The Observers

There were two tail length observers, Observer 1 (MW), PhD student, who was inexperienced with using the device, palpating joints, and assessing vulva cover, and Observer 2 (SB), a veterinarian, who was experienced with using the device, trialled on approximately 200 ewes prior to this study. Observer 1 was trained in all the measurement methods including BCS at the beginning of the trial by one member of the research team (CM). There were three assistants who had designated roles of data input, race assistant, and EID scanning/device assistant. Data input was performed using an electronic tag reader linked to a manual data entry tablet for tail length measures and condition score. Weight was recorded via electronic scale and linked via NLIS panel reader for each animal.

### 2.6. The Assessment

Five yearlings were assessed first and used to train Observer 1 in using the measurement methods and to compare the techniques and assessments of both observers. On the first day of the study, assessments began at 1020 h and concluded at 1800 h. Thirty-nine yearlings were assessed in an outdoor undercover race first with 5–7 yearlings in the race at a time. Due to weather conditions, 12 had to be assessed inside the adjoining woolshed, where 2–4 yearlings were in the race at a time. Weaners were all subsequently assessed inside the woolshed race with 4–6 weaners in the race at a time. Observer 1 always assessed the sheep first and observers swapped after assessing a race load of sheep. The observers, when not assessing tail length, moved away from the race and out of earshot to be blinded to the results. 

On Day 2, tail length measurements were repeated in the same order as Day 1. Assessments began at 0830 h and concluded at 1530 h. All sheep were weighed, and data were recorded electronically. All weaners were assessed in an outdoor undercover race first with 8 weaners in the race at a time. Twelve yearlings were assessed outside, and 39 were assessed inside the woolshed race due to weather conditions. 

### 2.7. Statistical Analyses

Statistical Packages for the Social Sciences (SPSS^®^ Release 27.0 2020 IBM^®^) was used for data analyses. The tail measurement methods produced binary (vulva cover, joints) and continuous (length) data. Observer and retest consistency analyses were performed using Cohen’s Kappa for binary data to assess agreement, Weighted Kappa for ordinal data to assess agreement, and Intraclass Correlation Coefficient (ICC Types used for observer and retest consistency respectively: single measure, 2-way random effects model using a consistency definition; single measure, 2-way mixed effects model using a consistency definition) for continuous data to assess reliability. 

Point biserial correlations were used to investigate the relationships between the length measurement with vulva cover and joint measurements. Fisher’s Exact Tests (FET) were used to assess association between vulva cover and joint measurements. T-tests were used to discern the significance of mean differences of length measurement between the vulva cover and joint measurements.

Landis and Koch’s (1977) [40] interpretations of the Kappa reliability coefficients were used in this study: Poor agreement (κ ≤ 0.40), moderate agreement (0.41 < κ < 0.60), substantial (0.61 < κ < 0.80), and almost perfect agreement (κ ≥ 0.81). The ICC coefficient interpretations of Koo and Li (2016) [41] were used in this study. The correlation interpretations of Cohen (2013) [42] were used, where coefficients ≥0.5 were classified as strong and coefficients between 0.3 and 0.49 were moderate. 

## 3. Results

### 3.1. Reliability

For both age groups, results indicate substantial to almost perfect observer and retest agreement for the vulva cover measurement (Table 2 and Table 3). The length measurement device on yearlings gave almost perfect observer consistency (Observer 1 vs. Observer 2), and substantial to almost perfect retest consistency (e.g., Observer 1 vs. Observer 1). For weaners, there was moderate (Day 1) and good (Day 2) observer consistency (Table 2), and moderate (Observer 1) and good (Observer 2) retest consistency (Table 3).

The joint palpation method provided the lowest observer consistency coefficients (Table 2 and Table 3). The highest agreement was on Day 2 for yearlings, while the remaining results were of poor agreement. Observer 2 had substantial retest agreement for yearlings, and Observer 1 had moderate observer agreement for weaners, the remaining results were poor agreement. The measurements of one observer were used for analysis of each method, based on the highest retest and observer consistency.

### 3.2. Yearlings

Yearling ewes weighed 46.7 kg on average (35.6–60 kg) and had an average body condition score of 2.9 (range: 2–3) from Observer 1’s data, based on highest reliability (Weighed Kappa statistic). For the analysis of all measurement methods of yearling tails, data from Observer 2 (Day 2) were used. The descriptive results for yearling tail measurement methods are presented in Table 4. There was a significant strong point biserial correlation (r_pb_ = 0.7, *p* < 0.01) between the yearling length measurement and the vulva cover measurements. The majority of the yearling tails assessed did not cover the vulva (72.5%). On average, tails classed as covering the vulva were significantly longer than tails classed as not covering the vulva by 16.2 mm on average t = 6.3 (df 48, *p* < 0.01). The shortest tail covering the vulva measured 41 mm and the longest tail not covering the vulva measured 62 mm. There was a significant association (FET, *p* < 0.01) between the joints and vulva measurement methods, where yearlings’ tails with only one joint were more likely to be classed as not covering the vulva than tails with two palpable joints. There was a significant moderate point biserial correlation (r_pb_ = 0.41, *p* < 0.01) between the yearling linear and the joint measurements. There was a significant difference in length between tails with one and two joints, where the yearling tails that had two joints were 15 mm longer on average than tails with one joint t = 5.8 (df 48, *p* < 0.01). 

### 3.3. Weaners 

Weaner ewes weighed 27.3 kg on average (range: 18.2–40 kg). For the analysis of weaner tail measurement methods, data were used from Observer 2 (Day 2) for the length measurements, Observer 1 (Day 2) for the joint palpation measurements, and Observer 1 (Day 1) for the vulva cover assessment. The descriptive results for weaner tail measurement methods are presented in Table 5. There was a significant strong point biserial correlation (r_pb_ = 0.5, *p* < 0.01) between the weaner length measurement and the vulva cover measurements. The majority of the weaner tails assessed covered the vulva (72.9%) and were significantly longer than tails that did not cover the vulva by 10.8 mm on average t = 4.0 (df = 45, *p* < 0.01). The shortest tail covering the vulva measured 48 mm and the longest tail not covering the vulva measured 71 mm. There was a non-significant association (FET, *p* = 0.059) between the weaner tail joints and vulva measurement methods. Tails that had two joints were more likely to be classed as covering the vulva; however, tails with one joint were relatively evenly classed as covering or not covering the vulva. There was a significant strong point biserial correlation (r_pb_ = 0.64, *p* < 0.01) between the weaner linear and the joint measurements. There was a significant difference in length between tails with two joints and one joint, where the weaner tails that had two joints were 8.3 mm longer than tails with one joint t = 3.0 (df = 45, *p* < 0.01).

## 4. Discussion

This study provides information on the reliability of three tail length measurement methods and their relationship in our sample of Merino yearling and weaner ewes. The most reliable methods of measurement were length and vulva coverage, and the least reliable was the number of palpable coccygeal joints. In this sample, yearling and weaner tails that covered the vulva were significantly longer than tails classed as not covering the vulva by 16.2 mm and 10.8 mm on average, respectively. It is an important finding to have two reliable methods for measuring sheep tails for valid assessments of tails which could be utilised for monitoring practice by producers and for research. The results of such assessments could then be compared against best practice recommendations and prompt change for improvement where tails are being docked shorter than recommended. Furthermore, these findings provide a basis for future research to be conducted in this area to refine these methods and increase clarity around how to communicate and implement the recommended tail length for the advancement of sheep welfare. 

Of the measurement methods used in this study, the length and vulva cover measurements were the most reliable between (Table 2) and within (Table 3) observers. From moderate to good observer and retest consistency was found using the length measurement and there was moderate–substantial observer agreement and substantial retest agreement using the vulva cover measurement. The observer consistency using the joint palpation method ranged from poor to moderate for yearlings and remained poor across both days for weaners. The retest consistency using the joint palpation method for Observer 1 for both yearlings and weaners was moderate, and for Observer 2 substantial for yearlings and poor for weaners. Of the two studies that also used a standardised tool for length measurement [27,28], only one reported reliability between observers’ lamb tail length measurements [27]. Goodwin et al. (2007) reported high inter-observer reliability demonstrating a strong correlation coefficient of r = 0.75 (*p* < 0.0001). However, after the initial reliability trial of observations at docking, reliability was not reported for observations at weaning or market age. Although Lewis (2013) did not perform reliability tests, it was specified that just one experienced technician was responsible for the docking procedure and the measurements, which may have been a contributing factor to their narrow range of measurement results across both age groups. Neither Graham, Johnstone, and Riches (1947) nor Goodwin et al. (2007) reported information on who or how many people conducted the length measurements or docking procedure across the experimental locations. Studies researching on-farm welfare measurements investigated the reliability of binary vulva cover as a tail length measurement method and found high percentage agreement [35] and poor to perfect [35] and fair to good [36] reliability using the Weighted Kappa statistic between observers. The instances of poor inter-observer reliability reported by Munoz et al. (2017) using the Kappa statistic was explained to possibly be due to the low number of recommended length tails observed in the sample at certain time points. Intra-reliability was also reported by Munoz et al. (2017) to range from poor to perfect agreement using the Weighted Kappa statistic. The observer consistency reliability in the present study increased on the second day of assessment for most methods in both age groups, likely due to increased practice of the measurements, particularly for Observer 1 who had no experience of measuring tails prior to the study. Observer and sheep height may have impacted length assessment. Observers were required to squat in the race behind the sheep during assessment, differences in observer height and height relative to the sheep height could have affected the level at which the device was held for length measurement and the eye level at which vulva cover would be seen. Therefore, it is possible individual observer physicality may impact the reliability and contribute to variation of the measurements. 

Each of the three measurement methods has unique advantages and challenges associated with implementation on-farm, outlined below in Table 6. For the assessment of Merino yearling and weaner ewe tail length, the results of this study indicate that of the three methods assessed, the most ideal methods are the combined use of the highly reliable length measurement and vulva cover methods. The vulva cover measurement is reflective of the guideline recommendation and already used as an on-farm animal-based welfare indicator [35,42], and the length measurement provides further information on the appearance and amount of tail remaining. Using these methods in conjunction would enable clear comparison to the best practice recommendations and, where tails are docked too short, insight into how much change is required to meet best practice guidelines. 

The number of palpable coccygeal joints observed in this study should be considered conservative underestimates due to the method of joint palpation. To be counted as intact, joints needed to be moveable and the subsequent vertebrae palpable to ~1cm below the joint. Tails that had been docked through the third joint space would have three coccygeal vertebrae where the third joint was docked through. In which case, in our study, these tails would have been classed as having two joints of the tail. Whereas in other research, at docking for example, tails docked through the third joint space were grouped as having three joints of the tail [33].

In this sample of merino ewes, tails that covered the vulva of yearlings and weaners measured 57.6 mm (range: 41–68 mm) (*n* = 14) and 63.7 mm (range: 48–79 mm) (*n* = 30) on average, respectively, and were associated with containing two coccygeal joints (more than two coccygeal vertebrae). Tails that did not cover the vulva of yearlings and weaners in the sample measured 41.3 mm (range: 29–62 mm) (*n* = 36) and 52.8 mm (range: 38–71 mm) (*n* = 17) on average, respectively, and had less than two coccygeal joints. 

Although it is difficult to make direct comparisons to previous research, these results do fit in with studies using similar measurement methods [11,27,28,33]. In previous studies, there are unclear methods of length measurement [11,33], inconsistencies in how tails were measured across experiments [11], different docking and measurement ages [11,33], the lack of [27,28] or retrospective [33] assessment of vulva cover, and the breeds and management of sheep differ greatly from that of the Australian industry [27,28]. This study’s weaner and yearling tail length measurements were conducted in a similar way to that of Goodwin et al. (2007) and Lewis (2013), where a device was used to standardise the length measurement and their approach led to the creation of the device used in our research. These previous studies were conducted in the United States of America, with either breeds not specified [27] or crossbred sheep [28], and were docked at younger ages compared to the sheep in our study. The weaner tails assessed in the current study were 22.68 mm longer [27] and 7.42 mm shorter [28] than the previous research using a standardised method of length measurement (Table 5). Our weaner results correspond more closely with the length measurements of Lewis (2013) than those of Goodwin et al. (2007). In comparison to an early Australian study, the average length of the weaner tails in our study was 2.8 mm shorter than measurements of 9-month-old Merino ewes with tails docked through the third joint space reported by O’Halloran et al. (1984) [33]. However, the sheep assessed by O’Halloran et al. (1984) were approximately 3–4 months older than our weaner ewes. 

Given that only 14 yearling ewes were assessed as having tails that covered the vulva in our study, the interpretation of results and comparisons to other studies are done so cautiously (Table 4). The yearling ewes’ tails assessed in this study that covered the vulva were 12.5 mm longer [27] and 13.9 mm shorter [33], on average, than studies using length measurements of sheep at similar ages that had tails docked at the caudal folds [27] or through the third joint space [33]. The majority of the yearling tails assessed in our study did not cover the vulva and measured 41.3 mm on average (Table 4). O’Halloran et al. (1984), reported similar results with the average length of ewe tails docked through the second joint space measured at 38.5 mm [33]. It is important to note that tail docking operators were different between the cohorts, potentially explaining some of the variation between and within the groups.

Monitoring tails after docking will provide information on whether the current docking length covers the vulva into adulthood, and on variation between sheep and operators. This information is important to enable assessment and adjustments towards best tail docking practice if tails are being docked shorter than recommended. In this study we did not measure tails at docking or follow the same sheep through the age groups, but it has been demonstrated that tails do increase in length after docking [27,28] and grow relative to the body [11]. Despite the aforementioned limited information on methodology, early Australian research indicates that lamb tails docked to partially cover or extend beyond the tip of the vulva maintain or achieve vulva coverage, on average [11]. Therefore, it is expected that tails that do not cover the vulva at ages after docking, including those in this study, would have been docked shorter than the length of the vulva as lambs. This provides evidence of the need for assessment of tail length after docking, so that docking practices can be evaluated, and corrected in the next season, if necessary, to ensure that the tail will be protective for the life of the sheep.

The many ways in which tail length is described could lead to many interpretations of the recommended length. Understanding how tail length descriptions relate to each other facilitates clear communication of how to dock tails at the best practice recommended length and how to monitor length over time. It is important that there is common language used regarding tail length that indicates tails should cover the vulva at docking and throughout the lifespan of the ewe and be a similar length in males. While several past studies have investigated tail length, there have been multiple and sometimes inconsistent ways of describing and/or measuring tail length. The methods used in this study were chosen as they are not only used in producer-facing publications by industry and government describing the recommended length but are also descriptors commonly used by producers, based on the authors’ previous research and experience with and in the industry. Some industry publications relate the recommended length of covering the vulva to leaving three coccygeal joints of the tail intact or docking through the third joint space. Detailed methods of joint palpation are not provided in past tail length research, and the method developed and used in this study had the pitfalls of excluding joints that had been docked through and poor reliability. The method of counting coccygeal joints and/or vertebrae could be ideal for use as a descriptor and measurement method across the sexes. A method not investigated in this study, but used in past research and sometimes in industry publications or resources, is the reference to the caudal fold attachment, and to dock past where the two folds distally attach to the tail. This was not possible to be investigated as we assessed tails after docking, so the caudal folds are either much more challenging to identify or, if docked above or through, are not present. This method may also present challenges at docking due to the individual anatomy of each sheep where each position of caudal folds may differ, and in some cases may be too short to cover the vulva even when kept intact. Another descriptor sometimes used is to leave the area of skin bare of wool underneath the tail intact, to cut beyond this point to achieve the recommended length. Similarly, this was not practically able to be investigated in this study. There remains research to be done to provide clarity and coherence between the different ways of describing tail length.

This study provides a basis that could be built on to further investigate the different tail length measurement methods and descriptions on a larger scale. Future research could involve refining the methods used in this study, to modify the length measurement using the device to further standardise the method, and to clarify a method for counting coccygeal joints and/or vertebrae. Once refined, testing the methods on a wider scale across commercial properties and involving producers would provide further insight into the practicality of the methods and how monitoring tail length could become a common on-farm practice. It would also be valuable to assess different measurement or descriptive methods at docking and following sheep through to maturity and after parturition to assess how each different method and description fits or differs from the recommended length at different ages and as the body changes.

## 5. Conclusions

The results of this study demonstrate that length and vulva cover assessment are reliable measures of tail length. The strength of the reliability of the measures between and within observers indicate that the device engineered for length measurement is effective, and confirms the vulva cover assessment remains relevant. These two methods provide reliable information that enable comparison to the recommended tail docking length which could assist with on-farm tail length monitoring by producers and/or researchers to ensure that tail docking practice results in the appropriate perineal protection for the lifetime of the sheep. Joint palpation was the least reliable method of tail measurement but is used as a length descriptor by industry and producers. The tails in this sample that were of the recommended length, covering the vulva, measured 10.8–16.2 mm longer than tails not covering the vulva, at 63.7 mm (weaners) and 57.6 mm (yearlings) on average, and contained more than two coccygeal vertebrae. Further investigations to refine the methods used in this study, and to assess tail length using all methods from docking through to maturity will highlight the practicality of on-farm use for monitoring and provide understanding as to the best way(s) to describe the recommended tail length at docking and older ages.

## Figures and Tables

**Figure 1 animals-13-00963-f001:**
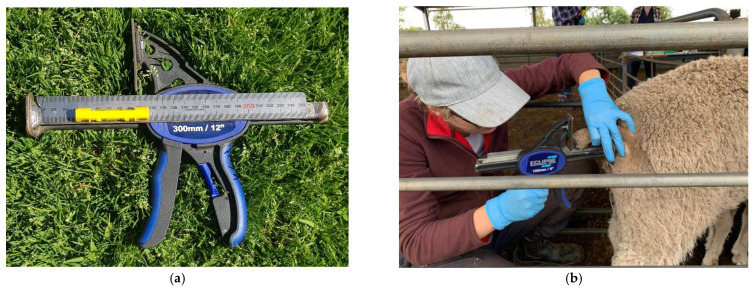
(**a**) The device created and used for the length measurement of tails. (**b**) The device in use on a weaner ewe.

**Table 1 animals-13-00963-t001:** Descriptions of the tail measurement methods used.

Measurement Method	Description
Vulva cover	Vulva cover was assessed on a binary scale. If the tail was equivalent to the length of, or longer than, the lower commissure of the vulva when held down with the sheep at rest it was scored ‘yes’, if shorter and did not cover the vulva it was scored ‘no’.
Joints	The joints of the tail were palpated from the first moveable coccygeal joint and were discretely recorded as 0/1/2/3 palpable joints remaining. A moveable joint was recognised if the subsequent coccygeal vertebrae could be felt 1 cm below the joint. That is, joints that were cut through were not included as being moveable joints.
Length	A device was created specifically for this study, engineered by combining a clamp with a steel ruler and bubble level fixed to the side (See Figure 1). The flat base of the clamp was placed underneath the sheep’s tail, resting underneath the caudal folds, against the anus of the sheep with the tail resting on top of the clamp in a horizontal position using the bubble level. The clamp was gently released by the observer to slide to the end of the tail stump, resulting in a measurement of the tail length in millimetres.

**Table 2 animals-13-00963-t002:** Observer consistency coefficients for each measurement method for yearlings (*n* = 51) and weaners (*n* = 48), significance at the *p* < 0.01 level indicated by **, significance at *p* < 0.05 level indicated by *.

Observer Consistency		Yearlings		Weaners	
Measure	Test	Day 1	Day 2	Day 1	Day 2
Length (mm) (95% CI)	ICC	0.8 ** (0.7–0.9)	0.9 ** (0.8–0.9) ^1^	0.7 ** (0.5–0.8) ^1^	0.8 ** (0.6–0.9) ^1^
Vulva cover	Cohen’s Kappa	0.6 **	0.8 **	0.6 **	0.5 **
Joints	Weighted Kappa	0.3 **	0.7 **	0.1	0.3 *

^1^ missing data point.

**Table 3 animals-13-00963-t003:** Retest consistency coefficients for each measurement method for (*n =* 51) and weaners (*n =* 48), significance at the *p* < 0.01 level indicated by **.

Retest Consistency		Yearlings		Weaners	
Measure	Test	Observer 1	Observer 2	Observer 1	Observer 2
Length (mm) (95% CI)	ICC	0.8 ** (0.7–0.9)	0.8 ** (0.7–0.9) ^1^	0.6 ** (0.4–0.8)	0.8 ** (0.6–0.9) ^2^
Vulva cover	Cohen’s Kappa	0.7 **	0.7 **	0.7 **	0.6 **
Joints	Weighted Kappa	0.5 **	0.7 **	0.5 **	0.1

^1^ missing data point. ^2^ 2 missing data points.

**Table 4 animals-13-00963-t004:** Descriptive statistics of the measurement methods used to assess yearling tails.

Measurement	Descriptive Statistic/Category	Vulva Cover		Joints	
		N	Y	1 *	2
Joints	*n*	37	14		
	1	34	1		
	2	3	13		
Length	*n*	36	14	34	16
	Range (mm)	29–62	41–68	29–60	39–68
	Mean (mm)	41.3	57.6	41.1	56.1
	Std. deviation (mm)	8.7	6.9	8.7	8.2

* Note: tails measured to have 0 joints in the tail (*n* = 4) that did not cover the vulva have been combined into the 1 joint category.

**Table 5 animals-13-00963-t005:** Descriptive statistics of the measurement methods used to assess weaner tails.

Measurement	Descriptive Statistic/Category	Vulva Cover		Joints	
		N	Y	1 *	2
Joints	*n*	13	35		
	1	9	13		
	2	4	22		
Length	*n*	17	30	22	25
	Range (mm)	38–71	48–79	38–79	50–78
	Mean (mm)	52.8	63.7	55.3	63.6
	Std. deviation (mm)	8.6	9	10.1	8.8

* Note: tails measured to have 0 joints in the tail (*n* = 3) that did not cover the vulva have been combined into the 1 joint category.

**Table 6 animals-13-00963-t006:** Identified pros and cons of each tail length measurement method used in this study.

Method	Pros	Cons	Most Appropriate for
Device	Reliable Indicates tail appearance.	~10 s per sheep Physical contact with the sheep, potential level of discomfort. Requires consistent pressure applied against the ischial tuberosities for consistent results. Manual data entry. However, potential for modification to be automated using NLIS. Potential observer variation due to physicality; differences may impact how the measurement is taken.	Research With modification and refined engineering could be useful for producers.
Vulva cover	Reliable Quick and simple to learn and perform. Lower commissure of the vulva easily identified. Non-invasive and involves minimal physical contact for the sheep.	Wool on the tail introduces potential for difficulty assessing vulva coverage. No indication of level of vulva cover from the binary data. Observer physicality may impact the point of view of the tail. Vulva size changes during pregnancy and after parturition could impact assessment of coverage.	Producers Valid if assessed from tail height.
Joints	This method could be useful for assessment and recommendations for both sexes, if refined.	Least reliable method. Difficult to learn and perform. Challenge of where to begin counting the moveable joints and whether to include joints that have been docked through as being intact joints.	Method refinement required.

## Data Availability

The data presented in this study are available on request from the corresponding author.

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
