# Peer review of "Measuring Sheep Tails: A Preliminary Study Using Length (Mm), Vulva Cover Assessment, and Number of Tail Joints"

_animals, 2023, doi:10.3390/ani13060963_

Round 1
Reviewer 1 Report
I am unsure about the journal Animals, but I have previously been taken to task over the word “hogget”, which is apparently not an internationally accepted term. Either use “yearling” or actual age “10 months” as in Table 1. This reviewer would particularly like to know the ages of the two groups in the methods. The lambing periods are shown but the date of day 1 of the assessment is not. Age would help further work, particularly after the large range in Table 1. This is born out by the discussion at lines 408 to 412.
The authors claim to have assessed body condition score, but then did not use it in the analysis? Very fat sheep are harder to find bones in than skinny weaners. Does the deposition of fat on the rump reduce tail length? Or are these weaners and hoggets too young to be fat or over fat?
I struggle with the contention that docking at the right length would improve animal welfare. The pain experienced by a tail docked too short would in my estimation be the same as a tail docked too long. Whether it is subjectively assessed or measured length they are both docked and both would hurt. I daresay they would by the same amount and I am in no way trying to encourage an experiment that would attempt to test that in anyway. The risk of dags, flystrike, skin cancer and prolapse are other things affected by length of tail but I would argue that not all animals with docked tails will suffer any or all of these and yet all will suffer docking. It would be good if the authors can state this clearly in the introduction and then clarify that it is the risk for short versus long in these other welfare compromises in their discussion at line 328. These risks remain and are not assessed in this experiment and to claim there might be some improvement of welfare in the current manuscript is speculative.
I also really struggle with why producers are going to want to monitor tail length. Once it is done you are not likely to change it. Yes, in learning for the following year, but research is the only likely use for monitoring.
The questions posed at line 116 are valid but there is repetition in adding: “In answering these questions, this study aimed to provide understanding into how each method relates to the others and the reliability and practicality of implementing each method on Merino hogget and weaner ewes. Therefore, this study aims to provide insight into which method(s) are most appropriate for measuring tail length on-farm, and to establish descriptions to enable interpretation of self-reported tail length” has some detail that should be covered in methods and some that should be deleted.
The terms inter-rater and intra-rater are used in this manuscript. They may well be terms that biometricians use but we only have two observers here and to accept this as inter-rater reliability is perhaps stretching it. Variation between two observers would be more accurate. Particularly at line 27 in the abstract – line 28 clarifies it and the sentence about raters could be left out.
If I can be blunt about the device for length measurement with a spirit level attached. It is not a surveyor’s instrument! This will be level with the ground in one direction when the bubble is centred, not with the spine of an animal that may kneel at the front or crouch at the back when you begin playing with its tail! The device from Goodwin et al. (2007) had sufficient backstop to ensure the tail was held as near possible to perpendicular to the rump and perhaps if not in line with the spine perhaps at a consistent angle to the rump?
Beyond line 422 up to 477 is speculation and literature review. Some of it has been noted earlier and is not necessary. The caudal folds are a great indicator until you realise how much you change those by the stretch and flex of the tail! The coccygeal vertebrae could be improved by comparing with scanning or carcase work but the number of vertebrae in the tail do not actually practically matter at all it is the length - whether two bones or three or two and a half.
The paragraph from line 76 to 91 is a list of publications which is relatively thorough. However, the paragraph has no start or end, there is no reason for why this list is provided. Table 1 also does this same thing.
Tables
Table 1 header says “… number of coccygeal joints, and relation to caudal fold attachment.” Which should probably say “… number of coccygeal joints (Joints).” The short form of “joints” is not declared in the header and there does not seem to be any clear caudal fold data in the table until the footnotes. Also the column (n) is last in the table but first in the header.
There is an extraordinary number of footnotes to Table 1. Surely the ages in footnote 3 could be summarised in the text more appropriately by “between such and such” or “up to 8 weeks”? This table needs a substantial rethink. It does not help the reader much in its current form. It is the kind of information one might see in a review paper but for this manuscript I am not convinced.
Table 5 is called descriptive statistics BUT there are no units and no description in the header. I assume range is in mm? Both the standard error of the mean and the standard deviation are provided – use SD and drop SED. Ditto for Table 6 (page line numbers have gone a bit crazy in this table)
Table 7 is unnecessary. These points should be discussed (briefly) in the discussion and not displayed in a cumbersome table
References
As a result of the following I would suggest the references need double-checking. Between lines 568 and 572 the references are not in an appropriate format.
Between lines 586 and 592 the references are not in an appropriate format.
Reference at line 603 not in appropriate format
Reference at line 609 and 610 are not in appropriate format
Specific small changes listed below.
Line 10 “… predominantly to prevent flystrike.” ‘reduce flystrike’ or ‘decrease flystrike’.
Line 68 “…between 24-54% of short-tailed sheep reported by…” Suggest a change of wording here. These are not short-tailed sheep but have been made to have short tails by docking.
Line 43 “… aid mating and parturition …” Sorry but I cannot have this message propagated in this fashion. It is wholly and solely the BELIEF that it would aid mating and parturition. Why did they not dock cattle and horses? It is also a mistaken belief. You will find that Scottish Blackface have their tails left long in the belief their udder does not get chilled – how then should we protect wild Bighorn sheep udders in much colder conditions than Scotland? “… in the belief it might aid mating and parturition…”
Line 70 “…some producer’s tail length descriptions…” It was more likely the description of the sheep tail lengths!
Line 96 “…interpretating…” interpreting
Line 98 “…methods can be utilised by producers and/or researchers on-farm.” Researcher maybe but producers?
Line 103 “… assessing pasture and nutritional quality…”
assessing pasture quantity and nutritional quality
Line 108 “…genetic quality for breeding” …superior genotypes for breeding”
Line 118 “on-farm?’.” Question mark fullstop
Line 159 “a Numnuts® applicator” The address of the supplier of this novel device is required. It is possible that for the journal Animals “elastrator rings” may not be adequate description either?
Reference to the infrared imaging at line 169 is really interesting. The greater detail in the section from line 178 to 183 is even more interesting and then the report at line 184 leaves the reader crushed. Delete all this if it is not in the paper.
Line 221, 222 and 224 “device linear measurements” is and odd term.
The information in tables 3 and 4 that some data points were missing is interesting but might be overkill. If the authors are enormously concerned decrease from n = 48 to n = 46 in the methods for example and remove those observations from the analysis.
At line 261 rpb should perhaps have “pb” as subscript to “r”?
The p values “p=0.000” in the section from lines 257 to 271 are astonishing.
I have a number of problems with sentence two of the discussion. It does not really say anything. Perhaps because it begins with “including” and originally began life as part of the previous sentence. It could say more conclusively that measured length (“linear length” is odd) and subjective vulva coverage were the most reliable methods of describing tail length.
Line 330 “within (Table 4Error! Reference source not found.) observers.” Is obvious.
Line 331 “Moderate-good…” “From moderate to good…”
Line 367 “in Table 7Error! Reference source not 367 found.” Again obvious
Line 376 “The number of palpable coccygeal joints observed in this…” should probably be a new paragraph
Lines 387 to 392 are very confident statements when the authors have just finished saying palpating coccygeal vertebrae is not that reliable.
The discussion at Lines 413 onwards got to something we really need to know and that is change with age. These are two different cohorts and in hindsight we cannot say they were docked in the same fashion.
Line 422 “marking” should be “docking” as in the rest of the document
Reviewer 2 Report
The manuscript deals with research aimed at investigating the reliability of three methods (linear length, vulva coverage, and joint palpation) for measuring the tail length of docked ewes. To this end, the inter- and intra-rater reliability of the three methods applied on 51 hogget and 48 weaner Merino ewes is compared.
This topic is of great interest not only because tail docking is a mutilation that causes pain to animals, but also because it is applied worldwide in sheep farming. For these reasons, the manuscript may be useful for improving sheep welfare.
However, the research design seems more suitable for a pilot study than for a research article, as the sample of animals analysed is too small.
The criterion adopted for the sample size does not seem appropriate because it derives from a welfare assessment protocol (AWIN) that has already been considered and weighted which factors influence the variability of the different parameters to be assessed. Indeed, the numerosity of some subgroups (bitches with vulva coverage Yes and with 2 joints; weaner with vulva coverage No) is very small (14, 16, 13 respectively). A power analysis approach could be useful and appropriate to define the sample size.
In any case, it is unclear in the study design which minimum difference in measurement made by each of the three methods tested is to be considered significant and which criterion was used to establish this (biological, ethical, regulatory, or other?).
If the authors intend to give their research the weight and force necessary to credit the results with greater scientific value and real impact on practice, then they should significantly increase the number of subjects analysed.
Otherwise, the manuscript should be revised and resubmitted as a pilot study proposing a methodological approach to be validated in subsequent and more extensive research to be conducted on other sheep breeds as well.
Reviewer 3 Report
General comments
This is an interesting study where the authors compared the inter- and intra-rater reliability of three different tail length measurement methods: vulva cover assessment, linear length and joint palpation. It is very well written and comprehensive in terms of literature reviewed. It should add to the existing information on where to dock in terms of length of the remaining tail, in countries where it is necessary to dock due to flystrike. Still, and in order to highlight the practicality of on-farm use that is helpful for the producers at the moment of docking, it remains necessary to assess tail length at the moment of docking and then follow the sheep from docking through to maturity. Both the length of the tail and the dimensions of the vulva will change with growth, and in relation to the whole body (i.e. different breeds) hence there is still research to do on this in order to improve the docking length recommendations for producers. Although the authors comment on this at the end of the manuscript, this is a very important point because at the moment of docking the linear measurement will probably become less important than the vulva cover. Hence this issue needs further research, so that docking practices can be evaluated and corrected in the next season, if necessary, to ensure welfare and that the tail length will be protective for the life of the sheep.
Specific comments:
I suggest that the section on infrared images could be deleted, because there are no results shown in this study and there is no point in mentioning this in M and M.
L330-331…between (Table 3) and within (Table 4Error! Reference source not found.) observers. Moderate-good inter- and intra-rater reliability was found…PLEASE CHECK.
Same problem at L 366-367 …..below in Table 7Error! Reference source not found PLEASE CHECK
Round 2
Reviewer 2 Report
The authors have addressed the observations I have made regarding the sample size and I agree with them.
However, it remains a fact that the research was carried out on a small number of animals, with very small subgroups, of a single breed (Merino), and in a farming context (Australia) very different from other parts of the western world (USA and Europe) where tail docking is carried out on animals of different ages and breeds. The authors themselves agree that 'There is a need for further research in this area, and it will be useful to include a variety of breeds to increase the worldwide applicability of the results'.
For these reasons, I reiterate that the manuscript should clearly state, also in the title and keywords, that this is a pilot, preliminary study. The current wording of the title is a bit misleading.
Author Response
The title of the manuscript has been amended to:
'Measuring sheep tails: A preliminary study using length (mm), vulva cover assessment and number of tail joints'
